# Prefoldin-5 Expression Is Elevated in Eutopic and Ectopic Endometriotic Epithelium and Modulates Endometriotic Epithelial Cell Proliferation and Migration In Vitro

**DOI:** 10.3390/ijms25042390

**Published:** 2024-02-17

**Authors:** Warren B. Nothnick, Wei Cui, Tommaso Falcone, Amanda Graham

**Affiliations:** 1Department of Cell Biology and Physiology, University of Kansas Medical Center, Kansas City, KS 66160, USA; agraham@kumc.edu; 2Department of Obstetrics and Gynecology, University of Kansas Medical Center, Kansas City, KS 66160, USA; 3Department of Cancer Biology, University of Kansas Medical Center, Kansas City, KS 66160, USA; 4Institute for Reproductive and Developmental Sciences, Center for Reproductive Sciences, University of Kansas Medical Center, Kansas City, KS 66160, USA; wcui@kumc.edu; 5Department of Pathology and Laboratory Medicine, University of Kansas Medical Center, Kansas City, KS 66160, USA; 6Section of Reproductive Endocrinology and Infertility—Obstetrics and Gynecology Institute, Cleveland Clinic Lerner College of Medicine, Cleveland, OH 44195, USA; falcont@ccf.org

**Keywords:** endometriosis, prefoldin-5, proliferation, migration, cell survival

## Abstract

Endometriosis is a common disease among women of reproductive age in which endometrial tissue grows in ectopic localizations, primarily within the pelvic cavity. These ectopic “lesions” grow as well as migrate and invade underlying tissues. Despite the prevalence of the disease, an understanding of factors that contribute to these cellular attributes remains poorly understood. Prefoldin-5 (PFDN5) has been associated with both aberrant cell proliferation and migration, but a potential role in endometriosis is unknown. As such, the purpose of this study was to examine PFDN5 expression in endometriotic tissue. PFDN5 mRNA and protein were examined in ectopic (lesion) and eutopic endometrial tissue from women with endometriosis and in eutopic endometrium from those without endometriosis using qRT-PCR and immunohistochemistry, respectively, while function of PFDN5 in vitro was evaluated using cell count and migration assays. PFDN5 mRNA and protein were expressed in eutopic and ectopic endometrial tissue, predominantly in the glandular epithelium, but not in endometrium from control subjects. Expression of both mRNA and protein was variable among endometriotic eutopic and ectopic endometrial tissue but showed an overall net increase. Knockdown of PFDN5 by siRNA transfection of endometriotic epithelial 12Z cells was associated with reduced cell proliferation/survival and migration. PFDN5 is expressed in eutopic and ectopic glandular epithelium and may play a role in proliferation and migration of these cells contributing to disease pathophysiology.

## 1. Introduction

Endometriosis is a disease common to women of reproductive age in which endometrial glands and stroma establish in the peritoneal cavity [1,2]. Characterized by chief complaints of infertility and pelvic pain, endometriosis impacts approximately 10% of women in their reproductive years [1,2]. Endometriosis is a complex and multifaceted disease, thought to occur due to abnormal hormonal, epigenetic, genetic, and immunologic/inflammatory pathways [3]. For endometriosis to establish ectopically, it is essential for the ectopic endometriotic lesions to survive, proliferate, migrate, and invade underlying tissues within the peritoneal cavity. In these respects, endometriosis shares many characteristics with cancer. Similar to the origins of endometriotic lesions, endometrial cancer originates in the endometrium and development and progression of both diseases is associated with estrogen exposure [4]. Further, both diseases are associated with misexpression of the transcription factor, c-MYC, which has been reported to be constitutively and aberrantly overexpressed in nearly three-quarters of human cancer [5], where it is proposed to drive cellular proliferation, migration, and invasion. The activity of c-MYC is controlled via complex mechanisms which include many c-MYC binding proteins, including prefoldin-5 (PFDN5) which has been proposed as a tumor suppressor [6,7]. Additionally, PFDN5 protein levels were found to be elevated in non-small-cell lung tumors and associated with overall survival rate [8], suggesting a potential role for this prefoldin in cancer pathophysiology.

While several studies have reported upregulation of c-MYC in the ectopic and eutopic endometrium of patients with endometriosis [9,10,11,12,13], PFDN5 expression and potential function remains to be evaluated in this disease. As PFDN5 could influence proliferation and/or migration of endometriotic lesions, the objective of the current study was to evaluate PFDN5 expression in endometriotic lesion tissue and evaluate functional consequences of modulating its expression in vitro on cellular events conducive to endometriosis pathophysiology.

## 2. Results

### 2.1. Endometriotic Lesion Expression of PFDN5 Transcript

We first assessed individual lesion *PFDN5v1* (herein referred to as *PFDN5*) expression expressed as a fold change from matched eutopic endometrium for each lesion. Of the 14 lesions obtained from the proliferative stage of the menstrual cycle, 11 expressed a higher level of *PFDN5* mRNA expression (>1.0-fold) compared to corresponding matched eutopic endometrium, while 3 expressed lower (<1.0-fold) levels compared to matched eutopic endometrium. For secretory stage specimens, 16 of the 19 expressed greater *PFDN5* transcript levels, while 3 expressed lower levels, compared to matched eutopic endometrial tissue (Figure 1A). Overall, proliferative endometriotic lesion specimens exhibited a net 1.53-fold increase in *PFDN5* transcript levels (*p* < 0.05; Figure 1B), while secretory stage lesion specimens exhibited a significant (*p* < 0.001) net 1.98-fold increase in *PFDN5* expression levels (Figure 1B). Comparing the level of expression of *PFDN5* transcript between proliferative and secretory stage lesions (fold change from matched eutopic) revealed no difference based upon stage of menstrual cycle (Figure 1B; *p* > 0.05).

### 2.2. PFDN5 Protein Localization in Control Eutopic, Endometriotic Eutopic, and Ectopic Tissues

PFDN5 protein expression was evaluated by immunohistochemistry on control endometrial biopsies (from women without signs/symptoms of endometriosis) as well as on endometrial biopsies from women with endometriosis and their matched ectopic lesion tissue (Figure 2). Like PFDN5 mRNA levels, protein localization of PFDN5 did not differ between proliferative or secretory stages of the menstrual cycle. During the proliferative phase, eutopic endometrial tissue from controls subjects showed no staining for PFDN5 (Figure 2A). In contrast, in the proliferative stage, eutopic endometrial tissue from subjects with endometriosis PFDN5 was robustly expressed in endometrial glands with some stromal staining (Figure 2B), and this glandular expression was also evident in ectopic glands (Figure 2C,C’). Like the proliferative stage control endometrial tissue, secretory stage eutopic endometrium did not express detectable levels of PFDN5 (Figure 2D). Secretory stage eutopic endometrial tissue from women with endometriosis expressed robust levels primarily in glandular epithelium with some stromal staining. PFDN5 was also detected in ectopic glands during the secretory stage (Figure 2F,F’). PFDN5 localization in eutopic and ectopic glands was predominantly cytoplasmic, with some nuclear localization (see Figure 2C’,F’; Table 1). Figure 3 shows box and whisker plots for proliferative (Figure 3A) and secretory stages (Figure 3B) of the menstrual cycle. Interestingly, in proliferative stage tissues, PFDN5 was predominantly expressed in the cytoplasm of the glandular epithelium of eutopic endometrium which became predominant in the epithelial cell nucleus of matched ectopic lesion tissue in 5/8 cases (Table 1). In contrast, this pattern of expression was lost in secretory stage specimens, where expression in the eutopic endometrial glands was still detected in 6/10 samples but expression in matched ectopic lesion tissue was inconsistent in localization with respect to nuclear versus cytoplasmic (Table 1).

### 2.3. PFDN5 Deficiency Is Associated with Reduced Cell Survival and Migration

To begin to examine the potential biological role of PFDN5 in endometriotic epithelial cells, we utilized the well-characterized 12Z cell line [14,15]. As PFDN5 has been shown to augment cell migration, coupled with the fact that cell migration is a characteristic of endometriosis, we first evaluated the impact of PFDN5 knockdown on cell migration. The 12Z cells transfected with *NT* siRNA occupied the majority of the gap (90.5% ± 4%; (mean ± SD); upper panel) 24 h post gap induction. In contrast, *PFDN5* siRNA-transfected cells migrated at a significantly lower rate compared to cells transected with *NT* siRNA (Figure 4 lower panel), occupying only 51.6% ± 11% of the wound area, which was significantly less (*p* < 0.001) compared to *NT* siRNA-transfected cells.

Next, we evaluated if PFDN5 plays a role in endometriotic epithelial cell survival/proliferation by transfecting 12Z cells with small interfering RNAs to *PFDN5*. As depicted in Figure 5, siRNA knockdown of PFDN5 was associated with a 54.3% decrease in cell survival at 48 h after transfection compared to controls. Collectively, we interpret these results to suggest that PFDN5 contributes to both cell survival/proliferation and migratory capacity in endometriotic epithelial 12Z cells.

### 2.4. PFDN5 Expression Positively Correlates with Markers of Cell Proliferation

To support the in vitro data, which suggested that PFDN5 may promote cell proliferation/cell survival, we examined if there was a correlation between *PFDN5v1* transcript expression levels with that of the known endometriotic epithelial cell proliferative/survival factor, RPLP1 [14]. Levels of endometriotic lesion *PFDN5v1* and *RPLP1* mRNA (expressed as delta ct values (*18S*–*PFND5* or 18S–*RPLP1*)) exhibited a Pearson correlation coefficient of 0.5906 with a significance value of *p* = 0.0003, considered highly significant.

## 3. Discussion

Endometriosis is a disease in which ectopic glands and stromal tissue establish and grow within the peritoneal cavity. The ectopic lesion tissue is associated with enhanced cell proliferation, migration, and epithelial to mesenchymal transition (EMT) during which epithelial cells lose polarity and cell-to-cell contacts, undergo remodeling of the cytoskeleton, and acquire migratory abilities and a mesenchymal-like gene expression program. The EMT process is proposed to play a role in the pathophysiology of endometriosis [16,17]. Overexpression of c-Myc induces EMT [18] as well as immune evasion, angiogenesis, ECM remodeling, cell migration and invasion [19], all which are associated with endometriosis pathophysiology. As mentioned earlier, c-Myc has been shown to be overexpressed in endometriotic lesion tissue [9,10,11,12,13], but PFDN5, which is capable of modulating c-Myc signaling, has not been assessed to date.

PFDN5 was first described by Mori and colleagues [20] as a novel c-MYC-binding protein (MM-1) using yeast two-hybrid screening of an HeLa cell cDNA library, primarily repressing the activation of E-box-dependent c-MYC transcription. A follow-up study by this same group [21] revealed that PFDN5 (MM-1) was expressed concordantly with c-MYC during G1 and S phases of the cell cycle and re-expressed after the G2 phase. This, in turn, might affect the functions of PFDN5 (MM-1), which may be dependent on c-MYC as well as may be c-MYC-independent. Further, in this study, it was revealed that an Ala to Arg substitution at amino acid 157 was frequently found in tissue culture cells and patient cells from leukemia, lymphoma, and tongue cancer patients. This mutation resulted in a loss of c-MYC binding function and tumor suppressor potential. This observation may explain why some reports suggest that elevated PFDN5 (MM-1) expression suppresses cell proliferation and migration compared to others, which have reported a stimulatory effect of PFDN5 (MM-1) on these cellular events.

In the current study, knockdown of PFDN5 using siRNA transfection resulted in reductions in both cell migration and cell proliferation. *PFND5* mRNA levels in endometriotic 12Z cells are rather abundant (average ct values = 20.64) and knockdown (to average ct values of 23.7) resulted in approximately 50–60% reduction in both cell migration and proliferation. These data may suggest that while PFDN5 appears to play a role in mediating these cellular events, PFDN5 alone does not solely regulate them in endometriotic 12Z cells in vitro.

In our study, PFDN5 protein was expressed by both eutopic and ectopic endometriotic epithelial cells, albeit at variable levels. Of note was the shift from predominantly cytoplasmic PFDN5 localization in epithelial glands of the eutopic endometrium (especially during the proliferative stage of the menstrual cycle) to predominantly nuclear expression in ectopic endometrial glands (during the proliferative stage of the menstrual cycle). All (8/8) eutopic endometrial tissue from women with endometriosis expressed PFND5 which was localized to the cytoplasm, with none of them expressing nuclear PFDN5. In matched ectopic lesion tissue, PFDN5 expression was predominantly nuclear (8/10), with only two solely expressing cytoplasmic staining. As mentioned earlier, an Ala to Arg substitution at amino acid 157 could result in a loss of c-MYC binding function and tumor suppressor potential, favoring cell proliferation. This postulate is supported by our in vitro data (Figure 5) and correlative data (Figure 6). On the other hand, cytoplasmic expression of PFDN5, which was predominant in the eutopic endometrium, might be suggestive of a role in cell migratory ability, which is supported also by our in vitro migration assay data (Figure 4) as well as by a recent study [22]. More specifically, overexpression of MM1/PFDN5 in human cell lines for mammary epithelial cell (MCF-10A), human type Ⅱ alveolar epithelial cells (A549), human tongue squamous carcinoma cells (UM1), and human keratinocytes (HaCat) resulted in enhanced cell migration of all cell lines, and this involved facilitating filopodia formation. As such, eutopic endometrial epithelial cell cytoplasmic expression might suggest a similar function. Luo and colleagues recently reported [23] that rapamycin-induced shorting of endometrial CRL-7566 (ovarian endometriotic cyst epithelium) cell filopodia was associated with reduced cell invasiveness/migration in scratch/wound assays. Eutopic endometrium from subjects with endometriosis is more invasive compared to that of controls [24] and it is postulated that this augmented invasiveness may promote the establishment of retrogradely shed endometrial tissue in the peritoneal cavity into ectopic lesions. Based upon our observations, PFDN5 may also be an additional factor which could promote lesion establishment of retrogradely shed endometrial tissue within the peritoneal cavity via such mechanisms. Together, these observations might suggest that in the context of endometriosis, elevated expression of PFDN5 may be capable of mediating pro-migratory and pro-proliferative effects in this tissue.

In contrast to eutopic and ectopic endometrial tissue from subjects with endometriosis, eutopic endometrium from control subjects expressed low/undetectable levels of PFDN5 protein. To the best of our knowledge, PFDN5 has not been examined in endometrial/uterine tissue of any species. *PFDN5* transcript (assessed by Northern blot analysis) was reported to be most abundant in the human pancreas with weak to faint expression in the kidney, skeletal muscle, placenta, liver, and lung [20]. Thus, PFDN5 may play a minimal role in the epithelial cell proliferation and/or migration that occurs during the menstrual cycle, and/or its expression may be at very low levels. From a diagnostic standpoint, assessment of PFDN5 immunolocalization in eutopic endometrium from cisgender females exhibiting signs/symptoms of endometriosis may provide a less invasive diagnostic tool, but larger studies would be required to provide sensitivity and specificity data. Nonetheless, initial studies are encouraging and support a role of PFDN5 in events conducive to endometriosis establishment and/or progression.

In summary, PFDN5 protein levels are elevated in eutopic and ectopic glandular epithelium of endometrial tissue from women with endometriosis. Expression within eutopic tissue is predominantly cytoplasmic, with nuclear expression becoming more predominant in matched ectopic endometriotic lesion tissue. Increased expression of PFDN5 may play a role in modulating cell migration and proliferation and, as such, play a role in the pathophysiology of endometriosis, but the mechanism by which it may do so remains to be determined.

## 4. Materials and Methods

### 4.1. Human Subjects and Tissue Acquisition

The study was approved by the institutional review boards of both the University of Kansas Medical Center and Cleveland Clinic. Written informed consent was obtained prior to surgical removal of endometriotic lesion tissue and endometrial biopsies. Women with endometriosis who presented with pelvic pain due to failed previous endometriosis treatment and were undergoing surgical removal of endometriotic lesion tissue were enrolled. No subjects had taken GnRH analogs or hormonal therapy within 3 months prior to surgery. For assessment of *PFDN5* mRNA expression studies, a total of 30 subjects (ages 21 to 45) were enrolled, from which 33 red, peritoneal endometriotic lesions were obtained. In 3 of these patients, we were able to obtain 2 separate lesions from different sites on the peritoneum. Endometriosis was classified as stage I/II or stage III/IV according to the revised (1996) American Society for Reproductive Medicine guidelines [25]. Of the stage I/II subjects (N = 12), five were in the proliferative stage of the menstrual cycle and seven were in the secretory stage, while in the remaining eighteen subjects, endometriosis was classified as stage III/IV, with nine in the proliferative and twelve in the secretory stage of the menstrual cycle. As in previous studies, stage of endometriosis had no impact on PFDN5 mRNA expression, and mRNA expression levels were classified by stage of menstrual cycle for all endometriosis specimens regardless of stage.

For immunohistochemical localization studies, archived tissue sections were retrieved from the Department of Pathology and Laboratory Medicine at the University of Kansas Medical Center. Controls consisted of patients without evidence of endometriosis, adenomyosis, fibroids, or endometrial carcinoma and displayed normal appearing endometrial tissue (N = 19 (N = 10 proliferative and N = 9 secretory stage of the menstrual cycle)). These controls consisted of subjects diagnosed with uterine polyps (N = 16), two cases for hysterectomy for non-gynecologic diseases (N = 2), and one for fertility assessment (N = 1). Patient ages ranged from 26 to 52 years of age with all exhibiting normal menstrual cycles. A total of eighteen (N = 18; (N = 8 proliferative and N = 10 secretory stages of the menstrual cycle)) endometriotic lesion and matched eutopic specimens were obtained for assessment, with patients ranging in age from 25 to 48 years of age. All tissues (controls and cases) were obtained from formalin-fixed, paraffin-embedded blocks which were fixed and processed in a similar fashion. For control and endometriosis specimens, the stage of menstrual cycle was determined by the date of the subject’s last menstrual period for premenopausal subjects.

### 4.2. Immunohistochemistry

Archived tissues were obtained from the Department of Pathology and Laboratory Medicine at the University of Kansas Medical Center for endometriosis and control patients. Tissues were fixed with 10% neutral buffered formalin and subjected to antigen retrieval. The slides were then dehydrated and rehydrated following standard procedures and subjected to immunohistochemical (IHC) localization using PFDN5 (Anti-PFDN5 antibody; Abcam 129116; Abcam, Cambridge, MA, USA) at a dilution of 1:100) as previously described [14,26]. IHC was performed following the recommendations of the manufacturer using a VectaStain ABC system (Vector Laboratories, Inc., Burlingame, CA, USA), and sections were counterstained with hematoxylin (Vector Laboratories, H-3404-100). Protein localization was identified as dark brown coloring on the tissue slides. Negative controls consisted of slides in which the primary antibody was omitted and replaced with an isotype-matched antibody. PFDN5 expression was quantitated using the semiquantitative H-score index previously described by us [14,26].

### 4.3. Cell Culture and Cell Transfection

The endometriotic epithelial cell line, 12Z, was obtained from Dr. Linda Griffith (Massachusetts Institute of Technology, Cambridge, MA, USA). Cell culture was conducted following the general approach as previously described [14]. Briefly, cells were cultured in phenol red-free Dulbecco’s Minimum essential medium (DMEM)/Ham’s F12 (Thermo Fisher Scientific, Waltham, MA, USA)) + 10% charcoal stripped FBS (Atlanta Biologicals, Atlanta, GA, USA) + 1% *v*/*v* Pen-Strep (Life Technologies, Carlsbad, CA, USA; defined herein as complete media) in T75 flasks and seeded at 1.5 × 10^6^ cells/flask in 10 mL of media until approximately 90% confluency. Cells were then passed and subjected to reverse-transfection using siPORT™ *NeoFX*™ transfection agent (ThermoFisher Scientific, Waltham, MA, USA) as described below.

ON-TARGETplus™ siRNAs (SMARTpool format) for *PFDN5* (L-011352-00) and a nontargeting pool (NT) negative control (D-001810-10) were obtained from Horizon Discovery (Lafayette, CO, USA). siRNAs were separately mixed with siPORT™ *NeoFX*™ transfection agent following the protocol supplied by Thermo Fisher Scientific diluted with DMEM/F12 media and added to culture wells for assessing siRNA knockdown (6-well plate), cell slides, or ibidi 2-well plates. The specified concentration of 12Z cells was then added to the siRNA/transfection reagent mixture and cells were cultured for either 48 h or 72 h post-transfection, after which *PFDN5* transcript expression and specific endpoints assessed for cell viability/survival and cell migration were assessed as described below. siRNA knockdown of PFDN5 was verified by qRT-PCR and Western blot analysis and resulted in greater than 80% knockdown of mRNA and protein.

### 4.4. RNA Isolation and qRT-PCR

Quantitative real-time PCR (qRT–PCR) was performed as previously described by us [14,26]. Briefly, total RNA was isolated using Tri-Reagent (Sigma Chemical Co., St. Louis, MO, USA) according to recommendations of the manufacturer. Total RNA (1 µg in 20 µL) was reverse-transcribed using reverse-transcription (RT) kits (Applied Biosystems; Foster City, CA, USA) following the manufacturer’s protocol. Primers for prefoldin-5, variant 1 (*PFDN5v1*), which is the predominant variant, were designed using Primer-Blast and synthesized by Integrated DNA Technology (IDT, Coralville, IA, USA): human *PFDN5v1* (NM_002624.3): forward, 5′-ACGTCCATTGCTCAGCTCAA-3′ and reverse, 5′-TCTTTCCCCTCGTTGCTCTT-3′. Primer sequences for *RPLP1* were previously reported [14]. Resulting material was then used for independent qRT–PCR. qRT–PCR was carried out on an Applied Biosystems HT7900 Sequence Detector. To account for differences in starting material, human 18S primers (cata-log #4310893E; ThermoFisher Scientific, Waltham, MA, USA) were used.

### 4.5. Cell Count/Viability Assessment

The 12Z cells (15,000/well in 6-well plates) were reverse-transfected with *PFDN5* or *NT* siRNA (30 nM final concentration) and cultured as described under “Cell culture and cell transfection” for 72 h. Media were removed from all wells, and cells were washed twice with room-temperature (22 °C) HBSS, after which HBSS was discarded. Five hundred microliters of 0.05% (*v*/*v*) trypsin-EDTA (0.53 mM *w*/*v*; catalog #25-052-CI; Corning, Glendale, AZ, USA) were then added to each well until cells became detached. Complete media (1.5 mL) were then added to each well, and contents were aspirated and centrifuged at 300× *g* for 3 min at 22 °C. Supernatant was discarded and cell pellets were resuspended in 1 mL of complete media. A 25 µL aliquot of resuspended cells was added to 5 µL Trypan blue and brought up to a final volume of 100 µL with HBSS. From this, 10 µL were placed into each chamber (2 total) of a hemacytometer. Cell concentration was calculated using the formula of average number of cells/field × 4 (dilution factor) × 10,000 and reported as viable cells/mL.

### 4.6. Wound Healing/Cell Migration Assay

The 12Z cells (10,000/well in 2-well ibidi plates) were reverse-transfected with *PFDN5* or *NT* siRNA (30 nM final concentration) and cultured as described under “Cell culture and cell transfection” for 24 h in complete media. Media and rubber gaskets were removed and 2 mL of DMEM:F12 containing 2% (*v*/*v*) CS-FBS and pen/strep was added to each dish, then photographs were taken to document the gap between cells. Cells were subsequently examined and photographed 24 h after creation of the gap. The percentage of cells that migrated into the gap was calculated using ImageJ FIJI (version 2.1.0/1.53c) and data were reported as percent of gap occupied by cells at 24 h compared to 0 h (zero percent occupancy) following a similar approach reported by Suarez-Arnedo and colleagues [27].

### 4.7. Statistical Analysis

*PFDN5v1* mRNA levels and PFDN5 protein H-Scores were first separately assessed within stage of endometriosis (stage I/II vs. stage III/IV in endometriosis subjects) and among stage of menstrual cycle. As no significant differences among their expression could be attributed to stage of endometriosis, data were assessed as endometriosis compared to controls. Stage of menstrual cycle did influence data; as such, data were analyzed by cycle stage and all data were assessed as eutopic endometrial tissue compared to endometriotic lesion tissue. Statistical analysis was conducted using GraphPad Instat3.10 (San Diego, CA, USA). Human IHC data were first evaluated for normalcy of distribution using a Kolmogorov and Smirnov test and did not follow Gaussian distributions. As such, comparisons among the study groups were then made using the nonparametric one-way ANOVA equivalent Kruskal–Wallis test or the nonparametric *t*-test, the Mann–Whitney U-test. For cell proliferation and migration studies, comparisons between *NT*- and *PFDN5*-siRNA-transfected cells were made using unpaired *t*-tests. Assessment of the relationship between *PFDN5v1* and *RPLP1* transcript expression in endometriotic lesion tissue was assessed by Pearson correlation coefficient. Significance was set at *p* < 0.05 for all experiments.

## Figures and Tables

**Figure 1 ijms-25-02390-f001:**
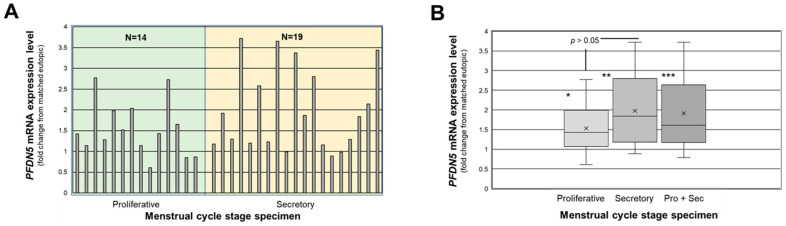
Fold change in *PFDN5* expression in red peritoneal endometriotic lesions. In (**A**), data are expressed as the fold change from eutopic endometrial levels for each lesion and were assessed separately by stage of menstrual cycle (proliferative and secretory) as well as combined across cycle stages (Pro + Sec). Data in (**B**) are displayed as box and whisker plots, where each data point from (**A**) was used to generate the graph. * = *p* < 0.05, ** = *p* < 0.001, *** = *p* < 0.001, respectively, comparing proliferative stage, secretory stage, and proliferative + secretory (Pro + Sec) ectopic lesion *PFDN5* levels to matched control eutopic endometrium levels. Fold change above eutopic *PFDN5* levels did not differ (*p* > 0.05) when comparing proliferative vs. secretory stages.

**Figure 2 ijms-25-02390-f002:**
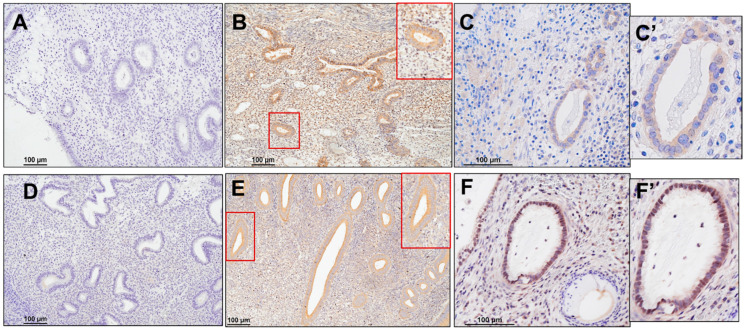
Immunohistochemical localization of PFDN5 in eutopic endometrium of women without endometriosis and in eutopic and ectopic endometrium from women with endometriosis. PFDN5 was localized in tissues, as described under “Materials and Methods”, using eutopic tissues from control (N = 20 total; N = 10 proliferative and N = 10 secretory stages of menstrual cycle) and eutopic and matched ectopic endometrial tissue from endometriosis subjects (N = 18; N = 8 proliferative and N = 10 secretory stages of menstrual cycle). Scale bar = 100 µm, magnification is 100× for panels (**A**–**D**) and 200× for (**C**,**F**). Red boxes within the right side of panel (**B**) and (**E**) are enlarged in the upper right hand corner of their respective panel (subpanel **B** and subpanel **E**) to highlight PFDN5 staining in the gland. (**C’**,**F’**) are further enlargement of the large glands in (**C**) and (**F**), respectively, and demonstrate cytoplasmic (**C’**) and nuclear (**F’**) PFDN5 localization.

**Figure 3 ijms-25-02390-f003:**
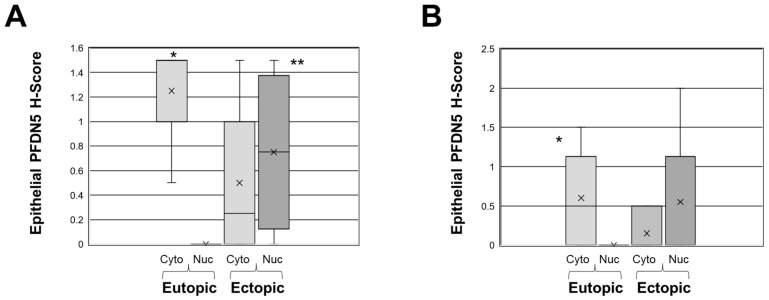
Box and whisker plots for PFDN5 H-Score in endometriotic ectopic and eutopic endometrial glandular epithelium. PFDN5 protein was localized in eutopic and matched ectopic endometriotic endometrial tissue in both proliferative (**A**) and secretory stage (**B**) specimens. In (**A**), * = *p* < 0.001 comparing eutopic cytoplasmic (Cyto) versus nuclear (Nuc) expression levels; ** = *p* < 0.05 comparing nuclear (Nuc) ectopic versus nuclear (Nuc) eutopic expression levels. In (**B**) * = *p* < 0.001 comparing cytoplasmic (Cyto) versus nuclear (Nuc) PFDN5 expression in eutopic endometriosis endometrium. Data were analyzed using Kruskal–Wallis nonparametric one-way ANOVA followed by Dunn’s multiple comparison test with planned comparisons.

**Figure 4 ijms-25-02390-f004:**
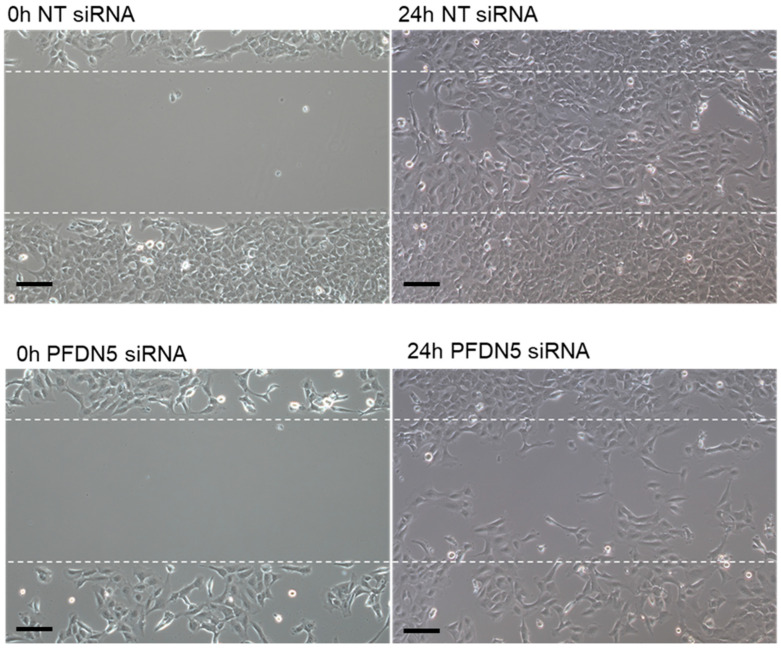
PFDN5 knockdown impairs endometriotic epithelial 12Z cell migration. The 12Z cells were reverse-transfected and plated into 2-well ibidi plates for 24 h, after which the gasket was removed to create a 500 µm gap between the cells. Migration into the gap was then assessed 24 h later. White dashed lines indicate gap area at 0 h after gasket removal and then at 24 h showing occupancy of migrated cells. Scale bar = 100 µm. Data are representative of 5 separate assessments per group (N = 5).

**Figure 5 ijms-25-02390-f005:**
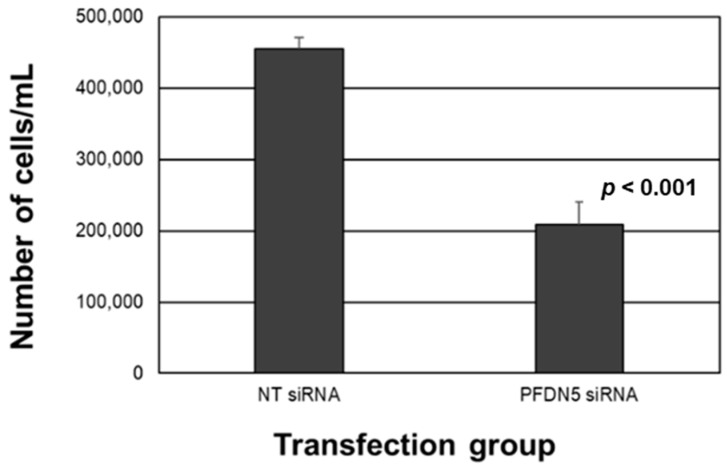
PFDN5 knockdown impairs endometriotic epithelial 12z cell survival in vitro. The 12Z cells were transfected with nontargeting (NT) siRNAs or siRNAs specific to PFDN5, and cell counts were assessed 72 h after transfection. Data were analyzed by unpaired *t*-test with equal variance (N = 4).

**Figure 6 ijms-25-02390-f006:**
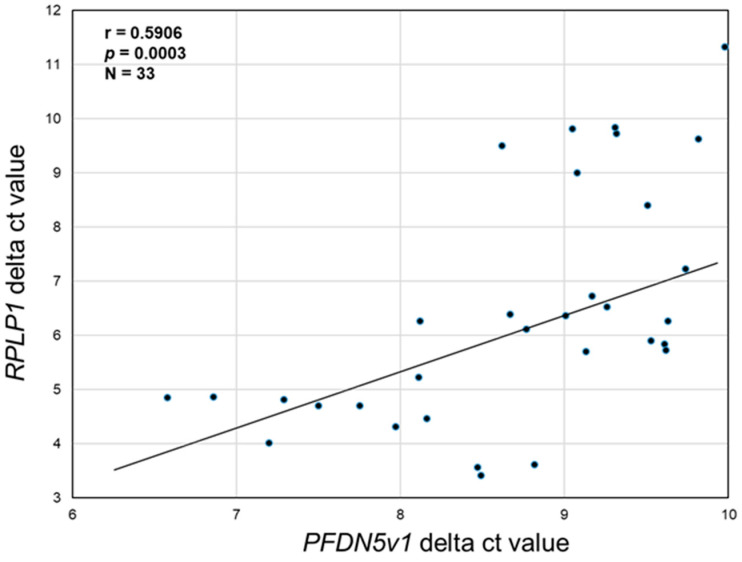
Correlation between endometriotic lesion tissue *PFDN5v1* and *RPLP1* transcript expression. cDNA from the 33 lesions (N = 33) used for *PFDN5v1* analysis (Figure 1) was assessed for *RPLP1* transcript expression, and delta ct values (*18S*–*PFDN5v1* or *RPLP1*) were calculated and used to examine the correlation between the two by calculating Pearson’s correlation coefficient.

**Table 1 ijms-25-02390-t001:** Level of PFDN5 and localization in endometriosis eutopic and endometriosis ectopic (lesion) glandular epithelial tissue during the proliferative and secretory stages ^1,2,3^.

Sample ID Proliferative Stage	Eutopic–Cyto	Eutopic–Nuc	Ectopic–Cyto	Ectopic–Nuc
0817368	1.5	0	0	1
1007838	1	0	0	1
1101676	1.5	0	0	1.5
2017EMP1	1.5	0	0	1.5
2017EMP3	1.5	0	1.5	0
2017EMP6	1	0	1	0.5
008732	0.5	0	0.5	0
2017EMP5	1.5	0	1	0.5
2017EMS2	1	0	0	2
2017EMS3	0	0	0	1.0
2017EMS4	1.5	0	0.5	0
2017EMS5	1	0	0	0
000084	0	0	0.5	0
009198	0	0	0	0
007838	0.5	0	0	0
032123	1.5	0	0	0
1013778	0	0	0.5	1.5
007204	0.5	0	0	1

^1^ Control endometrium expressed undetectable levels of PFDN5 under the described experimental conditions and data are not included in Table 1. ^2^ Stromal cell PFDN5 expression was variable and significantly lower compared to glandular epithelium expression, and data are not displayed in Table 1. ^3^ Abbreviations: Ge (glandular epithelium), Cyto (cytoplasmic staining), Nuc (nuclear staining).

## Data Availability

All data are contained within the article.

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
