# Peer review of "Prefoldin-5 Expression Is Elevated in Eutopic and Ectopic Endometriotic Epithelium and Modulates Endometriotic Epithelial Cell Proliferation and Migration In Vitro"

_ijms, 2024, doi:10.3390/ijms25042390_

Round 1

Reviewer 1 Report

Comments and Suggestions for Authors

Ref: ijms-2863003

Title: Prefoldin-5 Expression is Elevated in Eutopic and Ectopic Endometriotic Epithelium and Modulates Endometriotic Epithelial Cell Proliferation and Migration In Vitro

Recommendation: MINOR REVISION

Review:

The presented work delves into the intriguing topic of prefoldin-5 expression in both eutopic and ectopic endometriotic epithelium, exploring its influence on endometriotic epithelial cell proliferation and migration in vitro. Overall, it is a commendable piece of research, but a few minor shortcomings need attention.

The primary method employed for measuring expression, focusing solely on gene levels, may be limiting. Incorporating additional measures, such as protein expression, would provide a more comprehensive understanding of prefoldin-5's role in endometriosis. This could enhance the robustness of the findings and contribute to a more nuanced interpretation of the results.

The manuscript would benefit from including information about the effectiveness of gene silencing for PFDN5. Elaborating on the methodologies and outcomes of gene silencing experiments would strengthen the study's conclusions and offer valuable insights into the functional significance of prefoldin-5 in endometriosis.

A potential area for improvement lies in the succinct description of cell cultures. The brevity of these descriptions may pose challenges for researchers attempting to replicate the study. Expanding on the details of cell culture procedures, including specific conditions and techniques, would enhance the reproducibility of the experiments.

In conclusion, while the research is highly commendable with its engaging topic and significant findings, addressing these minor issues – incorporating additional measures of expression, providing details on gene silencing effectiveness, and expanding on cell culture descriptions – would further elevate the impact and clarity of the study.

Author Response

Uploaded as word file.

Reviewer 2 Report

Comments and Suggestions for Authors

The main question of the research is whether Prefoldin-5 expression differs in eutopic vs ectopic endometriotic epithelium, and whether this protein has any effect on the ectopic epithelium viability.

The topic is relevant to the field, as Prefoldin-5 has been associated with both aberrant cell proliferation
and migration but a potential role in endometriosis is unknown.

The research adds information on the Prefoldin-5 gene expression in endometriosis. Given moderate effects on cell viability and migration and a lack of action and regulation mechanisms, the research is a modest addition to the field.

The methodology is fine

The conclusions are fine

The references are appropriate

Author Response

Uploaded as word file.

Reviewer 3 Report

Comments and Suggestions for Authors

The idea of the study is interesting and the aim of the study is worth to be elucidated. Conducted laboratory tests are sufficient and well designed what makes study convincing and properly established. The study should be treated as a basic sciences experiment what is all good, however in regard to such often occurring disease being important clinical issue I would expect stronger clinical implications. Additionally in the introduction paragraph I would expand much more part presenting tested protein, as this is novel element in contrast to well known endometriosis. Moreover deeper explanation what is the rationale of such a choice would be advantage. In the references I found important positions, but I would expand this part slightly as I think there are missing some worth to be mentioned manuscripts. Finally, what is in my opinion important and sadly often occurs is logic mistake easy to be fixed - you use term “endometriotic tissue”… such tissue does not occur. It should be changed into endometriotic lesions, grafts, implants, etc, but for sure not tissue. As editorial mistake I count l. 22-23 “endometriotic tissue from women with and without endometriosis…” 

in general I recommend paper for publication in IJMS, pointed flaws are rather tiny and easy to be fixed so I can even suggest thick accept without changes having in a mind enlisted above “cosmetic” remarks. 

Author Response

Uploaded as word file.
